

# Synthesis and characterization of bismuth (III) complex with an EDTA-based phenylene ligand and its potential as anti-virulence agent

Melissa Beltran-Torres[1], Rocio Sugich-Miranda[2],
Hisila Santacruz-Ortega[1], Karla A. Lopez-Gastelum[1],
J. Fernando Ayala-Zavala[3], Fernando Rocha-Alonzo[2],
Enrique F. Velazquez-Contreras[1] and Francisco J. Vazquez-Armenta[2]

[1] Departamento de Investigación en Polímeros y Materiales, Universidad de Sonora, Hermosillo, Sonora, México
[2] Departamento de Ciencias Químico Biológicas, Universidad de Sonora, Hermosillo, Sonora, México
[3] Coordinación de Tecnología de Alimentos de Origen Vegetal, Centro de Investigación en Alimentación y Desarrollo, A.C., Hermosillo, Sonora, México

## ABSTRACT

A new acyclic bismuth complex (Bi-edtabz) was synthesized from a mixture of solutions of the ligand (EDTA-based phenylene) and bismuth under acidic conditions. Its anti-virulence properties were evaluated against *Escherichia coli* O157: H7, *Listeria monocytogenes*, *Pseudomonas aeruginosa*, *Salmonella* enterica sub. enterica serovar Typhimurium and *Staphylococcus aureus*. The bismuth complex was characterized by NMR, UV-Vis, FTIR, ESI/MS and TG. Furthermore, Bi-edtabz complex at 0.25–1 mM presented better antibiofilm properties against *E. coli* O157: H7 and *S. aureus* with values of biomass reduction of 30.1–57.1% and 37.8–55.5%, respectively, compared with the ligand edtabz. While biofilm formation of *L. monocytogenes*, *P. aeruginosa* and *Salmonella* Typhimurium was most impaired by edtabz (biomass reduction of 66.1–100%, 66.4–88.0% and 50.9–67.1%), respectively. Additionally, Bi-edtabz inhibited the swimming motility of *E. coli* O157: H7 (12.5%) and colony spread of *S. aureus* (47.2%) at 1 mM and inhibited violacein production, a *quorum-sensing* related pigment of the biosensor strain *Chromobacterium violaceum*. Hence, edtabz and the Bi-edtabz complex can be used as novel anti-virulence agents against pathogenic bacteria.

# INTRODUCTION

The increasing spread of antibiotic resistance has driven the search for new classes of antimicrobials. Metal complexes that have been used for years in the medical chemistry area have been largely ignored for antibiotic development (*Ndagi, Mhlongo & Soliman, 2017*). This class of compounds can adopt a wide range of three-dimensional conformations than other organic compounds leading to unique modes of action (*Bar,*

Corresponding author
Francisco J. Vazquez-Armenta,
franciscojavier.vazquez@unison.mx

*Pichon & Sutter, 2016*). These properties make them interesting starting points for the development of new antimicrobials (*Frei, 2020*).

Bismuth(III) is a borderline metal ion according to the acid-base theory of Pearson, but it has a high affinity for multidentate ligands containing O and N donor atoms (*Briand & Burford, 2000*). EDTA-based ligands have donor atoms that can coordinate with bismuth. It is well known that increasing the number of donor atoms of the ligands and the number of chelating rings usually results in higher stability of the complexes (*Beltran-Torres et al., 2019*). Nevertheless, this is not only the factor that regulates the metal coordination; other factors such as the preorganization, the charge of the ligand, the steric efficiency in which the ligand surrounds the Bi(III) ion to form a cage-like structure also play an important role (*Stavila et al., 2006*).

Several bismuth compounds have been synthesized for different purposes, from semiconductors to antimicrobials (*Marzano et al., 2013*). Bismuth coordination compounds inhibited the growth of pathogenic bacteria such as *Helicobacter pylori, Staphylococcus aureus*, *Escherichia coli, Pseudomonas aeruginosa*, *Salmonella* Typhimurium, *Shigella sonnet* and *S. dysenteriae* (*Vazquez-Munoz, Arellano-Jimenez & Lopez-Ribot, 2020*; *Wang et al., 2018*; *Marzano et al., 2013*). The antimicrobial mode of action of bismuth compounds is not yet completely elucidated. However, several studies indicated that this property is related to the interference with the cell wall synthesis, inhibition of ATP synthesis, and inhibition of key enzymes in the tricarboxylic acid cycle (*Wang et al., 2020*; *Marzano et al., 2013*).

However, the new trends in antibacterial agents are focused beyond their biocidal effect, and attention is being directed to their anti-virulence and pathogenicity activity. This approach includes the interruption of biofilm formation and intercellular communication, inhibiting toxin production and motility, necessary for the colonization of surfaces and tissues infection (*Defoirdt, 2018*; *Luna-Solorza et al., 2020*). Even when the antimicrobial activity of bismuth and its coordination complexes is recognized, their effects on bacterial virulence have not been evaluated.

In previous work, iron and copper complexes with an EDTA-based phenylene macrocycle (edtaod) and its open-chain derivative (edtabz) have been synthesized and evaluated as antibiofilm agents against food and clinical-related pathogens (*Vázquez-Armenta et al., 2021*). In general, the open-chain ligand edtabz and their metal complexes showed better anti-biofilm properties without biocide effects than the macrocycle ligand. Correlation analysis showed a positive relationship between molecular properties of compounds such as molecular weight, volume and the number of rotatable bonds and its antibiofilm activity (*Vázquez-Armenta et al., 2021*). Those results demonstrated the potential of coordination complexes in anti-virulence therapy to fight biofilm-related bacterial infections. Based on such findings, this study aimed to synthesize a bismuth(III) complex with an EDTA derivative ligand and characterize it through spectroscopic techniques. In addition, their effects on virulence factors of pathogenic bacteria such as biofilm formation, motility, and cell-to-cell communication were also investigated.

## MATERIALS AND METHODS

### Synthesis of the ligand

The edtabz ligand was synthesized as previously described by *Beltran-Torres et al. (2019)*. Specifically, 2.4 g (9.3 mmol) of EDTA dianhydride dissolved in 8 mL of dry dimethylformamide (DMF) was added to 2 mL (20 mmol) of aniline previously distilled. The resulting reaction mixture was left to stand overnight and any solids were filtered out. The filtrate was concentrated to 5 mL by a rotary evaporator, into which acetone was added. Precipitates formed were filtered off, washed with acetone several times until a colorless solid was obtained. Finally, it was vacuum-dried for 8 h at 25 °C. The purity was confirmed by IR and $^1$H NMR spectra and determining the decomposition point.

### Synthesis of the bismuth complex

The bismuth complex was synthesized by reaction of the appropriate nitrate salt and ligand. In 15 mL of water, 1 mmol of edtabz was suspended and solubilized by adding $Li_2CO_3$. This solution was mixed with 1 mmol of nitrate bismuth (10 mL aqueous solution, dissolved with a few drops of nitric acid), no visible change of color was observed. After stirring was continued for 30 min and the solution was heated at 40–50 °C. When it was concentrated to half the initial volume and stabilized to 25 °C, the product precipitated. A white powder was filtered off and vacuum-dried for 8 h at 25 °C. Yield: 0.4146 g, 54%. mp 238 °C (dec). Found: C, 33.18%; H, 3.12%; N, 10.49%. Calcd for $BiC_{22}H_{24}O_6N_4 \cdot 2NO_3$: C, 33.39%; H, 3.31%; N, 10.62%. Mass spectrum (ESI−) m/z: 709.6 (100%), [(M − H)−] (The mass spectrum of the complex is given in Fig. S1).

### Spectroscopic measurements

The bismuth complex was characterized in terms of $^1$H and $^{13}$C NMR, UV-Vis, Fourier-transform infrared (FTIR), mass spectra of electrospray ionization (ESI/MS) and thermogravimetric analysis (TGA) (*Beltran-Torres et al., 2019*). The $^1$H and $^{13}$C NMR spectra were obtained with a Bruker AVANCE 400 spectrometer for $D_2O$ solutions at 25 °C. The internal reference was sodium 2,2-dimethyl-2-silanpentane-5-sulfonate (DSS). IR spectra were recorded on a Perkin-Elmer FT-IR Spectrometer Model Frontier equipped with an ATR accessory. UV-Visible spectroscopy was carried out using Perkin-Elmer Lambda 20. For the pH-variable experiment of the Bi-edtabz mixture, the sample was dissolved in 0.1 M NaCl, and the pH values of the sample solutions were adjusted with 0.1 M HCl or 0.1 M NaOH to keep the ionic strength and sample concentration constant. A quartz cuvette of the spectrometer was loaded with the basic solution of the Bi-edtabz, and proper amounts of the acid complex solution were added to have a pH gradient of 0.5 in the final reading of the acid complex solution.

Mass spectra of electrospray ionization (ESI/MS) were obtained on 6,130 Quadrupole LC/MS of Agilent Technologies in the negative ionization mode. Thermogravimetric Analysis (TGA) was carried out on a thermogravimetric analyzer Perkin Elmer Pyris 1 TGA to study the complex's thermal stability and composition. Five mg of sample was set in a ceramic pan and analyzed at a 25–800 °C range under an $O_2$ atmosphere.

## Antimicrobial activity of ligand and bismuth complex

The antibacterial activity of edtabz and the bismuth complex was evaluated as previously reported by *Vázquez-Armenta et al. (2021)*. The tested pathogenic bacteria were *Escherichia coli* O157: H7 (ATCC 43895), *Listeria monocytogenes* (ATCC 7644), *Pseudomonas aeruginosa* (ATCC 10154), *Salmonella* enterica sub. enterica serovar Typhimurium (ATCC 14028) and *Staphylococcus aureus* (ATCC 6538). An inoculum of $1 \times 10^8$ CFU/mL for each bacterium was obtained from exponential phase cultures in nutrient broth (Luria-Bertani, Brain Heart Infusion or Tryptic Soy broths). Then, 5 μL of inoculum were added to a sterile 96-well microplate (Costar 96), followed by 295 μL of each compound at different concentrations (0–1 mM) diluted in the corresponding nutrient broth to achieve a final inoculum level of $1 \times 10^6$ CFU/mL. The microplate was incubated for 24 h at 37 °C. Bacterial growth in the presence of ligand or bismuth complex was inspected visually. The highest tested concentration (1 mM) did not inhibit the growth of the evaluated bacteria; thus, 0.25, 0.5 and 1 mM were selected for further analyze the anti-virulence activity.

## Effect of ligand and bismuth complex on biofilm formation of pathogenic bacteria

The capacity of bismuth complex and edtabz to prevent biofilm formation of pathogenic bacteria was evaluated by the crystal violet staining procedure as previously reported by *Vázquez-Armenta et al. (2021)*. Bacterial inoculum was prepared at $1 \times 10^8$ CFU/mL in their corresponding broth from an exponential phase culture. Two μL of the bacterial inoculum and 150 μL of each compound at different concentrations (0, 0.25, 0.5 and 1 mM) were taken and placed in sterile 96-well polystyrene microplates (Costar 96) and incubated for 24 h at 37 °C. After the incubation period, the culture was removed by aspiration, and the wells were washed three times with distilled water and let dry for 15 min. Subsequently, the formed biofilms were stained by adding 200 μL of 0.1% (w/v) crystal violet to each well (45 min at 25 °C). Then, wells were washed gently three times with distilled water to remove the unbound dye and dried for an additional 15 min. The bound dye to biofilms was solubilized by adding 200 μL of 20% acetic acid for 15 min. Finally, the optical density (OD) was measured at 580 nm in a FLUOstar Omega spectrophotometer (BMGLabtech, Chicago, IL, USA). Nutrient broth without any compound and bacteria was used as a blank, and the OD values were subtracted from treatment readings. Each experiment was carried out in triplicate, and results were expressed as percentages of inhibition compared with control samples (biofilms grown without compounds) following the formula:

$$\text{Percentage inhibition (\%)} = \frac{OD_{positive\ control} - OD_{treatment}}{OD_{positive\ control}} \times 100$$

## Effect of bismuth complex on swimming motility

The effect of bismuth complex and ligand on swimming motility of pathogenic bacteria was evaluated by exposing 10 μL of bacterial suspensions growth overnight (18 h, at 30 °C)

to the presence of 1 mM of each compound. The treated bacteria were placed in the center of Petri dishes that contained soft agar (0.3% agar), and untreated bacteria were used as controls. The Petri dishes were incubated at 30 °C, and the diameter (mm) of bacterial motility halos was measured after 24 h of incubation (*Bernal-Mercado et al., 2020*). Each experiment was performed in triplicate.

### Screening for anti-*quorum sensing* activity

The biomonitor strain *Chromobacterium violaceum* (ATCC 12472) was used for preliminary screening of anti-*quorum sensing* activity of the bismuth complex. For this purpose, the disk diffusion assay was employed where the inoculum of *C. violaceum* was grown aerobically in Lauria-Bertani (LB) broth at 30 °C for 18 h and adjusted to $1 \times 10^8$ CFU/mL. Then LB agar plates were spread with 100 µL of *C. violaceum* inoculum, and 20 µL of 2 mM bismuth complex or ligand solution was loaded to the sterile disks and placed on the surface of inoculated LB agar plates. The plates were incubated upright for 24 h at 30 °C, and a colorless appearance indicated the inhibition zone of QS (*Alvarez et al., 2014*).

### Experimental design and statistical analysis

A complete randomized design was performed, the evaluated factor was the concentration of compounds (0.25, 0.5 or 1 mM), and the response variable was biomass reduction of biofilms (%) and motility halos (mm). An analysis of variance (ANOVA) was carried out to estimate significant differences among treatments ($p \le 0.05$), and the Tukey-Kramer test was used for means comparison ($p \le 0.05$) in the software NCSS 2007.

## RESULTS

### $^1$H NMR spectra edtabz and Bi-edtabz

The edtabz ligand shows the typical derivative EDTA signals, Ha, Hb and Hc. Due to the symmetry of the ligand, only five signals in the $^1$H spectrum are seen (Fig. 1). Once the bismuth ion is coordinated, the equivalence of the aliphatic protons and the aromatic ones is lost. In Fig. 1A, protons from the aromatic ring in the free ligand are chemically equivalent, and just two signals for both the aniline rings are shown. Meanwhile, in the Bi-edtabz, the two aromatic rings can be distinguished (Fig. 1B). The $^{13}$C NMR spectra Bi-edtabz complex also confirms such a phenomenon. Nine carbon signals are observed in the $^{13}$C spectrum of the edtabz ligand (Fig. 2A), while seventeen signals are shown in the complex spectrum (Fig. 2B). To further understand the chemical environment, coordination mode of the edtabz with Bi$^{3+}$ and the proper protons assignment, a Heteronuclear Simple Quantum Coherence (HSQC) experiment was performed (Full HSQC spectrum is given in Fig. S2, and the integrated signals of the complex is given as well, Fig. S4).

In the y-axis, the aromatic carbons (Fig. 3), Cortho, Cmeta and Cpara are shown. Six signals can be seen, three for each aniline ring present in the ligand. In the x-axis, a series of multiplets are shown, which are difficult to assign, but the 2D experiment makes it easier. The multiplets that seem to appear as a doublet at ~7.5 ppm come from the

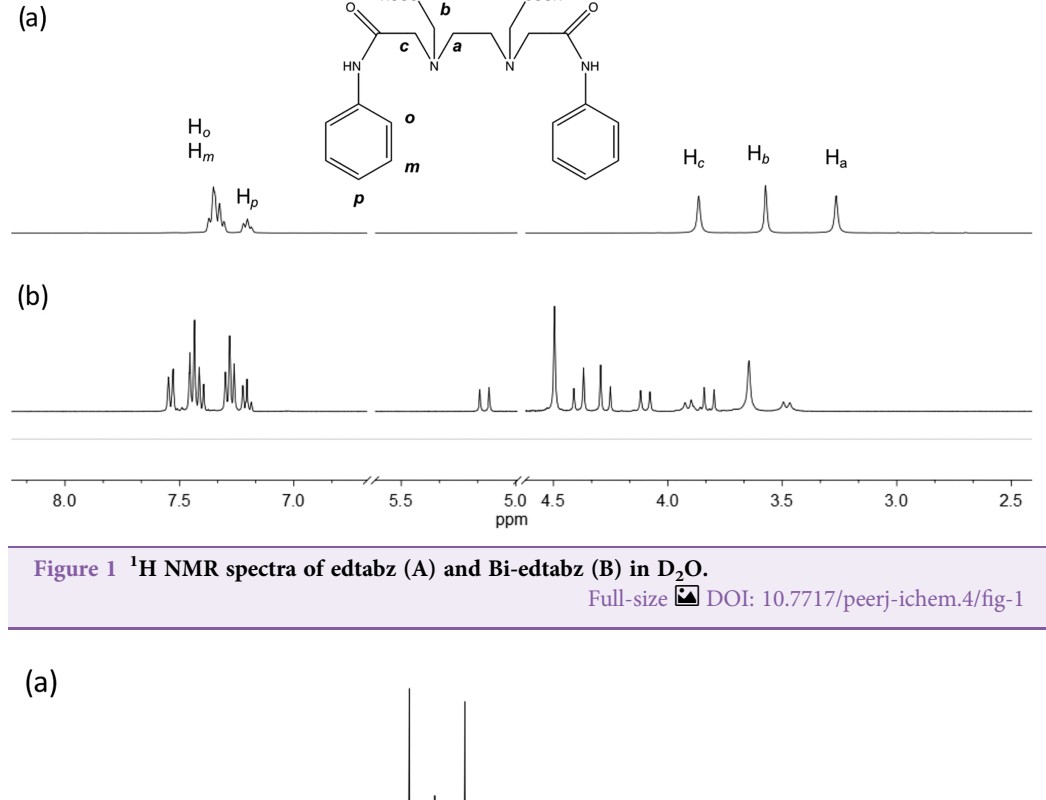

**Figure 1** ¹H NMR spectra of edtabz (A) and Bi-edtabz (B) in D₂O.

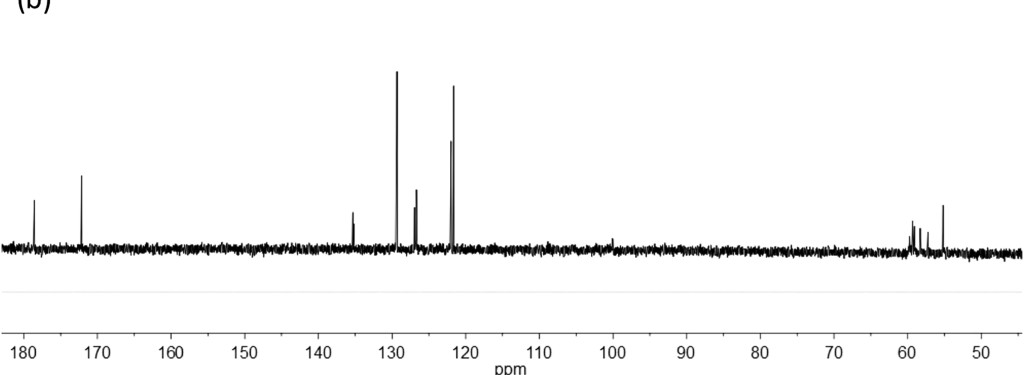

**Figure 2** ¹³C NMR spectra of edtabz (A) and Bi-edtabz (B) in D₂O.

aromatic protons in the ortho position. The multiplet at 7.43 ppm is a contribution of ortho as well as the meta protons. Following the dots in the HSQC experiment, the protons from para and meta form the multiplet at 7.28 ppm, and finally, the multiplet at 7.20 ppm comes from para protons.

The aliphatic protons can be observed in the graph's x-axis in the region from 5.2 to 3.5 ppm (Fig. 4). As previously mentioned, the number of signals observed, and the pattern

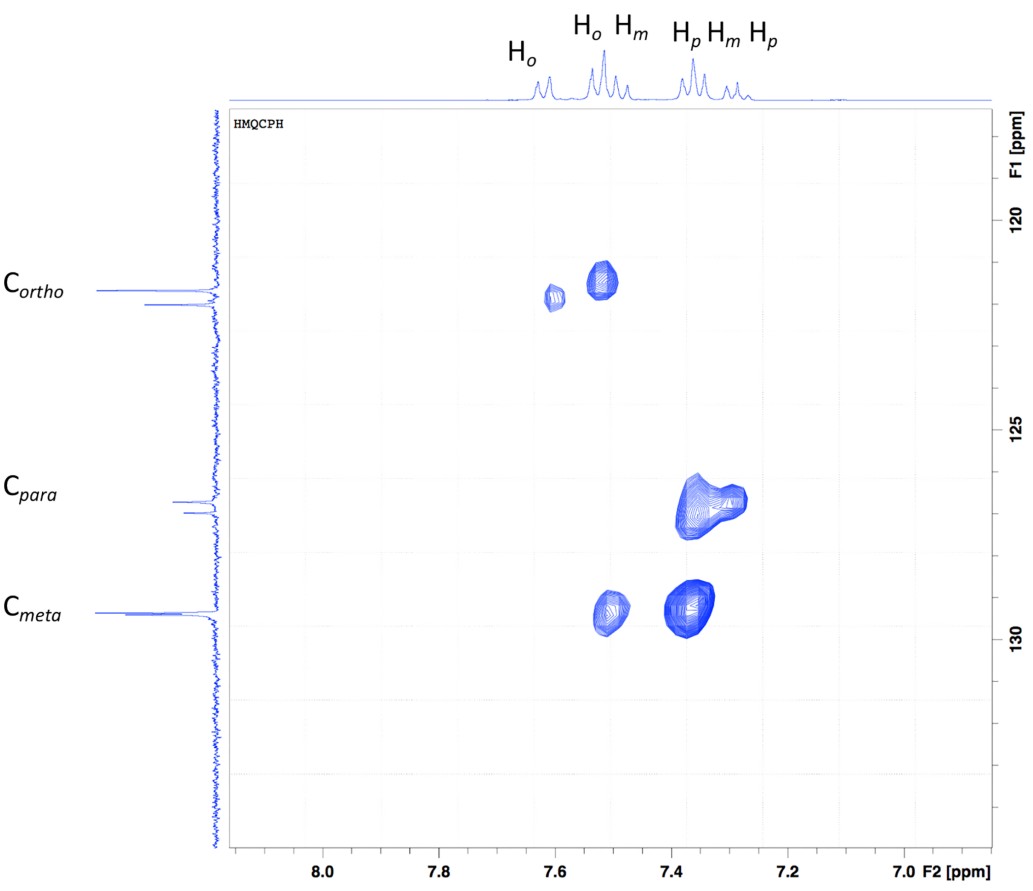

**Figure 3** HSQC of the Bi-edtabz in the aromatic region.

displayed can be attributed to a system that has gone from a flexible backbone to a rigid one. The proton at 3.65 and 4.50 ppm are assigned to Ha protons. The Hb protons are assigned to the signals at 4.29 and 4.37 ppm. These protons are sign to the methylene protons neighboring the carboxylate group in the acid; based on previous information, it can be speculated that one arm coordinates from above while the other coordinates from below or a side (*Beltran-Torres et al., 2019*). These protons are the ones that suffer the most from the complexation; hence the pattern of the doublet with uneven height can be seen. At 4.02, and 5.05 ppm, the doublets are attributed to Hc protons (J = 16 Hz). A doublet can be seen at 3.84 ppm with a J = 16 Hz, but no correlation in the spectrum can be seen. The small doublets signals at 3.86 and 3.41 ppm (J = 12 Hz) can be attributed to an equilibrium of species when instead of the two carbonyl groups are coordinating, only one is.

## Thermal analysis

The thermal stability of the Bi-edtabz was analyzed by TG analysis. The complex weight loss was monitored while the temperature increased at 10 °C min$^{-1}$ from 25 °C up to 800 °C, under an air atmosphere (Fig. 5). The thermal analysis of the bismuth complex

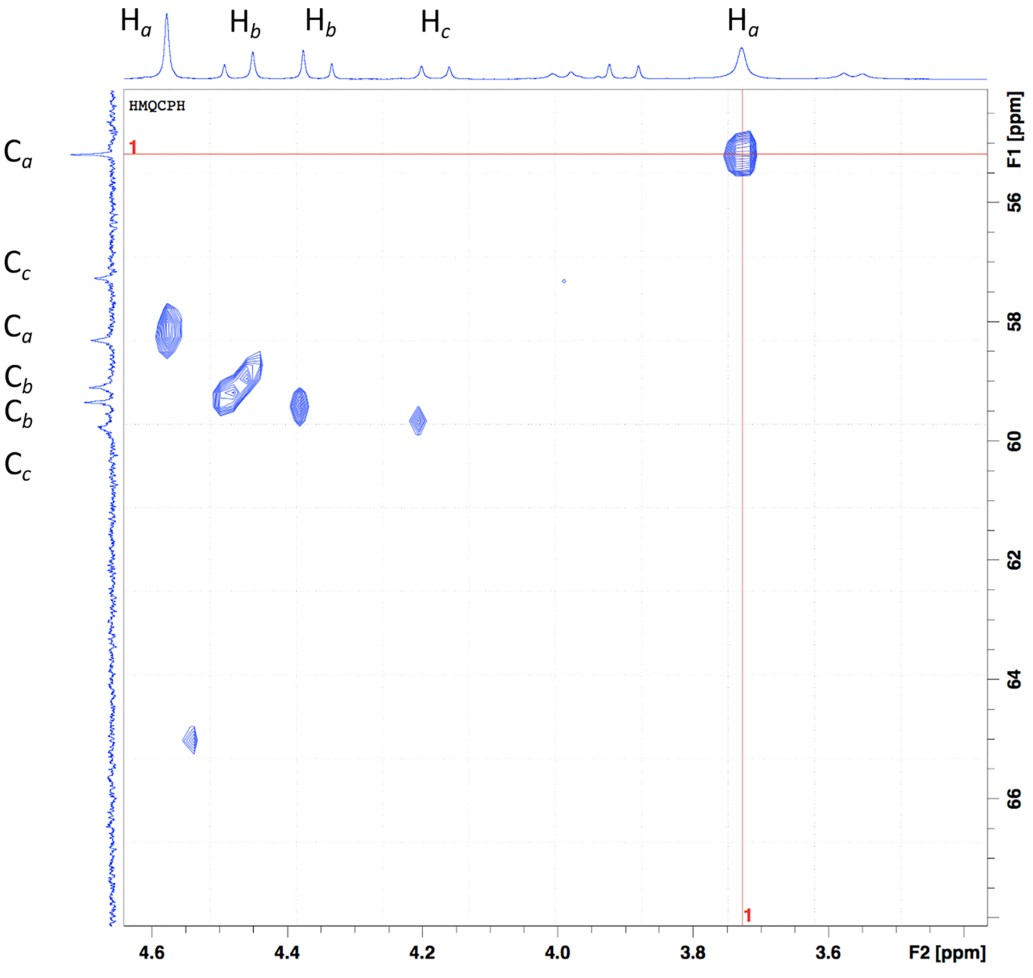

**Figure 4 HSQC of the Bi-edtabz in the aliphatic region.**

included four decompositions steps (180–562 °C). The first decomposition step (180–220 °C) involves a 7.4% mass loss due to the evolution of a nitrate molecule. The second decomposition step (220–242 °C) involves a loss mass of 7.4%, which could be attributed to another nitrate ion. This result indicates that one $NO_3$ molecule is part of the outer sphere of the complex (the nitrate molecule in the first decomposition step), and the other nitrate molecule is part of the inner sphere coordination; therefore, the second decomposition step is attributed to this nitrate molecule (*Taha et al., 2011*; *Radecka-Paryzek, Pospieszna-Markiewicz & Kubicki, 2007*; *Anjaneyulu, Prasad & Swamy, 2010*). The third and fourth stages of decomposition (242–374 °C and 374–562 °C) with 38.6% and 16.7% mass loss are attributed to the ligand evolution. The remaining 29% loss corresponds to bismuth residues.

## FTIR spectra of Bi-edtabz

The ligand and the complex's infrared spectrum were taken to understand further the structure of the complex and functional groups involved in the bismuth coordination.

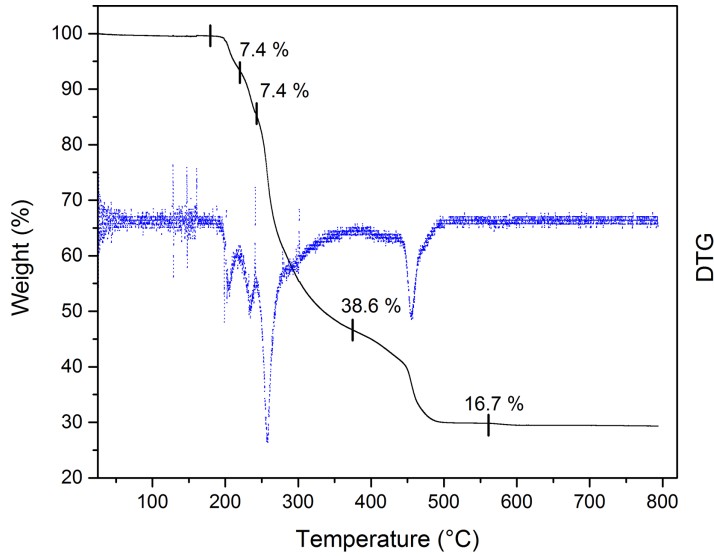

**Figure 5** TG curve of Bi-edtabz with a heating rate of 10 °C/min in an air atmosphere with a flow of 20 mL/min.

**Table 1 IR data of edtabzH$_2$ and Bi-edtabz.**

IR (cm$^{-1}$)

| Ligand/complex | $\nu$ N-H est | $\nu$ COOH | $\nu$ (CONH) (I) | $\nu$ (CONH) (II) | $\nu$ C-N |
|---|---|---|---|---|---|
| edtabzH$_2$ | 3,273 | 1,700 | 1,611 | 1,548 | 1,219 |
| Bi-edtabz | 3,291 | 1,620 | 1,591 | 1,570 | 1,296 |

Some informative bands of the ligand and its complex are given in Table 1. In the free ligand, the strong characteristic stretching band C=O of carboxylic acid appeared at 1,700 cm$^{-1}$ and moved to the lower wavenumber side by 80 cm$^{-1}$ upon complexation. This band shift indicates that the carboxylate groups participated in the formation of the complex. Other bands displaced to lower wavenumber sides upon complexation are the vibrational bands of amide C=O and C–N bonds, which indicates that the amide groups were also involved in the coordination metal ion (The FTIR spectra of bismuth complex and free ligand are given in Fig. S3).

## UV-Visible spectra of Bi-edtabz

At 242 nm, the edtabz ligand shows a band attributed to transition π–π* proper from the aromatic rings. In the bismuth complex, there are two bands, one attributed to the interaction of the ligand's aromatic rings, which shows a hyperchromic effect at 242 nm and 263 nm at a pH of 1.0 (Fig. 6A). The band at 263 nm is attributed to the metal-ligand transfer charge. The tendency of bismuth to form hydroxides prevented the attempt to titrate bismuth nitrate to ligand to corroborate the molar ratio of complex because the bismuth hydroxide precipitated as a white powder. In Fig. 6B, the pH-variable experiment is presented. At acid pH, the complex showed two shoulders, and as the pH became basic,

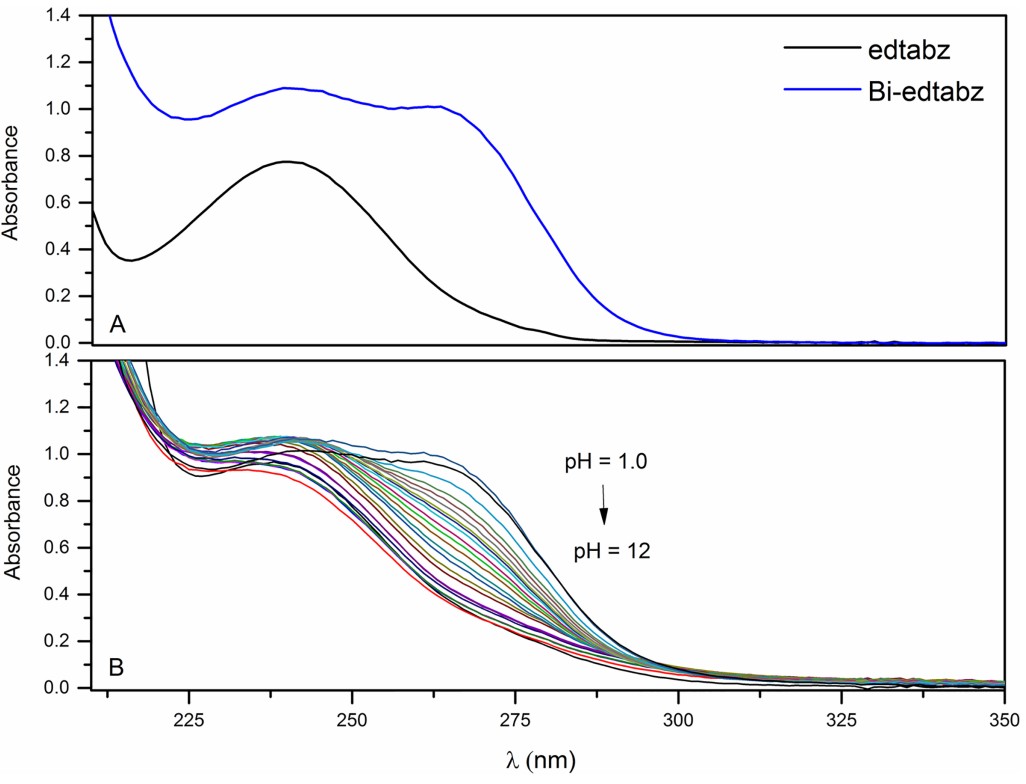

**Figure 6** UV-Vis spectrum of (A) edtabz and Bi-edtabz [0.04 mM] and (B) pH-variable experiment of Bi-edtabz [0.04 mM] in NaCl 0.1 M.

the metal-ligand transfer charge band decreased, and the spectrum at pH 12 is the same as the free ligand. A dissociation occurred, which could be to the formation of bismuth hydroxides.

## Effect of Bi-edtabz and edtabz on pathogenic bacteria in planktonic state and biofilm

Visual inspection of the bacterial cultures that grew in the presence of ligand or bismuth complex showed that both compounds at the highest concentration evaluated (1 mM) did not affect the viability of the bacterial cells. Thus, concentrations of 0.25, 0.5 and 1 mM were chosen to evaluate its effect on biofilm formation of pathogenic bacteria using the crystal violet assay where the amount of bound dye is proportional to the total biomass of biofilms (*Burton et al., 2007*). Figure 7 shows the percentages of biomass reduction of biofilms formed in the presence of edtabz or Bi-edtabz complex. Both compounds inhibited biofilm development in a dose-dependent manner and presented different inhibition patterns among bacterial species. Bi-edtabz complex at 1 mM showed higher biomass reduction against *E. coli* O157: H7 ($p = 0.0029$) and *S. aureus* ($p = 0.014$) (58% and 55%, respectively) than the ligand (43% and 37%, respectively). The ligand edtabz caused higher biomass reduction in *L. monocytogenes* ($p = 0.000067$), *P. aeruginosa* ($p = 0.0021$) and *Salmonella* Typhimurium ($p = 0.0409$) than Bi-edtabz complex. At 1 mM, edtabz

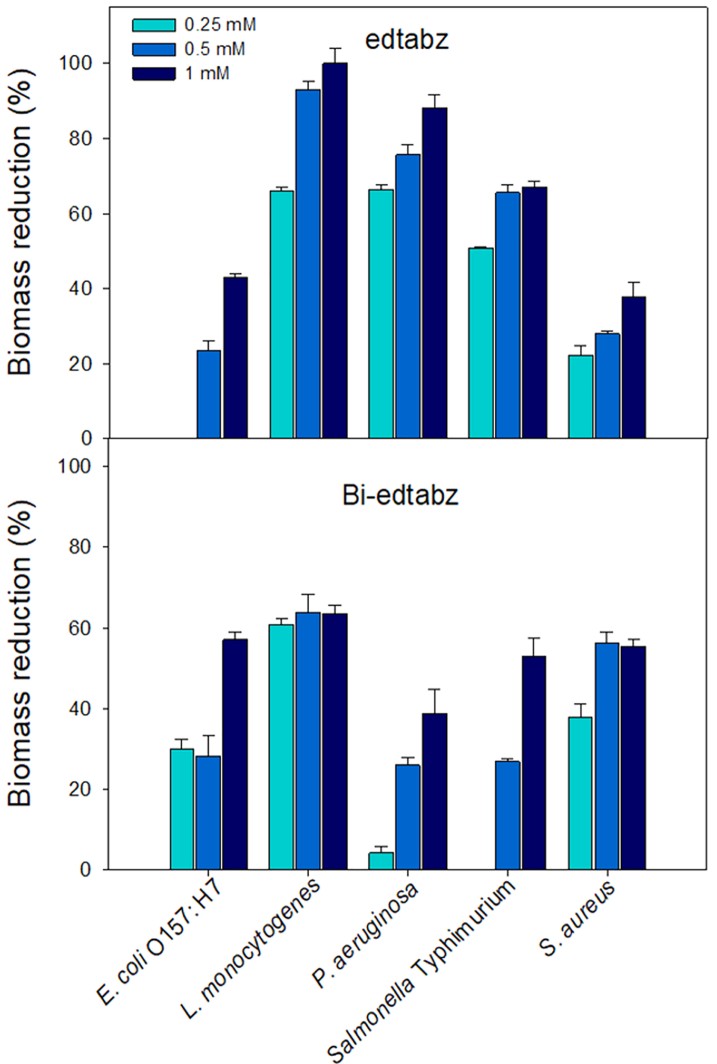

**Figure 7 Inhibition of biofilm formation of pathogenic bacteria incubated for 24 h at 37 °C in the presence of edtabz or Bi-edtabz determined by crystal violet assay.** Means for three independent experiments ± standard errors are illustrated ($p \leq 0.05$).

completely prevented biofilm formation of *L. monocytogenes* and reduced by 88% and 67% the biomass of *P. aeruginosa* and *Salmonella* Typhimurium biofilm.

## Effect of Bi-edtabz and edtabz on swimming motility

In Table 2, the effect of edtabz and Bi-edtabz on the swimming motility of pathogenic bacteria is presented. After 24 h of incubation at 30 °C untreated pathogenic bacteria displaced on the surface of soft agar showing motility zones from 40.9 to 78.1 mm. The presence of Bi-edtabz complex in the culture media impaired the motility of *E. coli* O157: H7 ($p = 0.00029$) and *S. aureus* ($p = 0.00001$) with reductions of 12.5% and 47.2%, respectively. While the motility of *L. monocytogenes*, *P. aeruginosa* and *Salmonella* Typhimurium was not affected ($p > 0.05$) by the bismuth complex. On the other hand,

**Table 2 Effect of edtabz and Bi-edtabz on swimming motility of pathogenic bacteria at 30 °C.**

| Bacteria | Motility zone (mm) | | |
|---|---|---|---|
| | Control | edtabz | Bi-edtabz |
| *E. coli* O157: H7 | 76.6 | 76.9 | 66.1* |
| *L. monocytogenes* | 40.9 | 41.0 | 41.0 |
| *P. aeruginosa* | 77.9 | 78.0 | 78.1 |
| *S. aureus* | 78.1 | 78.0 | 41.6* |
| *Salmonella* Typhimurium | 77.3 | 44.9* | 77.9 |

**Notes:**
Results are shown as mean ± standard error of three independent experiments.
* Difference between values in the same row ($p \leq 0.05$).

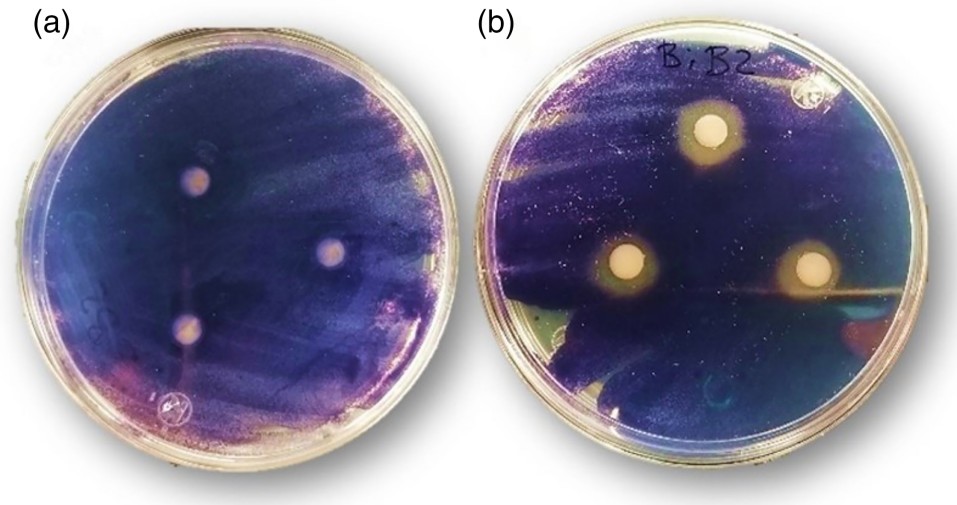

**Figure 8 Inhibition of violacein production by edtabz (A) and Bi-edtabz (B) in *C. violaceum*.**

edtabz only affected ($p = 0.00001$) the motility of *Salmonella* Typhimurium, showing 41.9% inhibition compared to untreated bacteria.

## Anti- *quorum sensing* activity of Bi-edtabz and edtbaz in *Chromobacterium violaceum* model

To explore the anti-*quorum sensing* activity of the bismuth complex as preliminary screening, biomonitor strain *C. violaceum* was used. This bacterial model produces a purple pigment (violacein) during the quorum-sensing activation; this production is regulated in response to self-produced acyl-homoserine lactones (AHL). Thereby, analyzing the pigment production in the bismuth complex presence, it is possible to evaluate the interference of a given substance in this process (*Alvarez et al., 2014*). As shown in Fig. 8B, Bi-edtabz inhibited the violacein production of *C. violaceum* as evidenced by the colorless halo (11.9 ± 0.4 mm), absent in edtabz treatment (Fig. 8A).

## DISCUSSION

In this study, a bismuth complex with an open-chained EDTA-based phenylene ligand was synthesized and characterized. Few studies of this type of compounds had been reported until date, specially the $^1$H NMR analysis. The diamagnetic property of bismuth made it possible to elucidate the coordination through this technique. As can be seen in Fig. 1, in the $^1$H spectrum of the Bi-edtabz many signals are presented. The number of signals increased in the complex, and some are seen as doublets of doublets; this response is characteristic of a rigid system. The edtabz ligand is known as a podand, this open structure provides flexibility to the ligand, which provides the facility to coordinate different metal ions because it is not restricted to a preorganized framework. Anyhow, once the ligand coordinates a metal ion, this open structure changes to chelate the metal. In this case, the chelation of the bismuth ion generates a rigid system. The bismuth ion presents an ionic radius of 103 pm ($Bi^{3+}$), which is considered big compared to other metal ions studied for this ligand ($Cu^{2+}$ and $Fe^{3+}$). The bismuth ion bulkiness comprises the ligand flexibility, and therefore almost all signals are seen doubled due to the loss of chemical equivalence of its protons.

In the pH-variable experiment, it was observed that a disassociation of the complex occurs as the pH becomes basic and the complex seems to be stable in acid media which can be a good property if the aim of this compound is to be use as a pharmaceutical. Therefore, the effect on biofilm formation, motility and cell-to-cell communication of pathogenic bacteria was evaluated. As shown in Fig. 7, edtabz and Bi-edtabz reduced the biomass of bacterial biofilms and this effect was dose- and species-dependent. $Bi^{3+}$ complexation resulted in an enhancement of antibiofilm activities of the edtabz ligand against *E. coli* O157: H7 and *S. aureus*. In a previous study $Cu^{2+}$ complexation improved inhibitory activities of edtabz against *S. aureus* while $Cu^{2+}$ and $Fe^{3+}$ complexation reduced the antibiofilm properties of edtabz against *E. coli* O157: H7 (*Vázquez-Armenta et al., 2021*).

In a metal complex formation a reduction of the polarity of the central metal atom occurs due to the sharing of its positive charge with the ligand; therefore, it favors the permeation of the complex through the lipid layer of the cell membrane (*Tümer et al., 1999*). In this case, the edtabz and Bi-edtabz do not present a direct correlation; thus, further investigation should be done to elucidate its antibiofilm mode of action. In this sense, studies focused on understanding the transition from planktonic free-living cells to complex bacterial communities in biofilms have been evidenced multiple factors involved in this process, such as the ability of bacteria to attach and colonize abiotic surfaces, the physicochemical properties of bacterial surface, the production of extracellular polymeric substances, and cell-to-cell communication (*Borges et al., 2016*; *Andrade et al., 2020*). These factors could be considered as starting points to get insight into the mode of action.

Biofilm formation is considered a virulence factor in some pathogenic bacteria since it helps establish infections. For example, *P. aeruginosa* can establish chronic biofilm-associated infections in patients with cystic fibrosis, where biofilm matrix confers resistance to antimicrobial treatments increasing mortality rate (*Harrington, Sweeney & Harrison, 2020*). In uropathogenic *E. coli*, biofilm formation keeps bacteria in the urinary

tract and hinders its eradication in catheter-associated infections (*Zhao et al., 2020*). In comparison, *S. aureus* can form biofilms in medical implants causing nosocomial bacteremia that are difficult and expensive to treat (*Garzón, Martinez & Molina, 2019*; *Oliveira et al., 2018*). Therefore, preventing the formation of biofilms is one of the strategies to combat bacterial infections, thus obtained results open the possibility of using edtabz and Bi-edtabz as selective inhibitors against the biofilm formation of pathogenic bacteria.

Another important virulence factor is bacterial motility; pathogenic bacteria use different types of movements mediated by flagella and pili, such as swimming, swarming, or twitching motilities involved in bacterial pathogenesis and biofilm formation. For this reason, bacterial motility is considered a common target in anti-virulence therapy (*Khan et al., 2019*). Swimming motility is defined as the individual movement of bacteria in liquid or low-viscosity surfaces without the need for biosurfactants (*Martínez et al., 2013*; *Ha, Kuchma & O'Toole, 2014*). Despite the recognized antimicrobial activity of bismuth-containing compounds (*Duffin, Werrett & Andrews, 2020*), few reports evaluated the effect on virulence factors of pathogenic bacteria, such as bacterial motility (*Alipour et al., 2010*). Thus, results obtained in the present study showed that the bismuth complex effectively interferes with this virulence factor in *E. coli* O157: H7 and *S. aureus*. In contrast, edtabz was effective only against *Salmonella* Typhimurium while the bismuth complex did not interfere with bacterial motility in this bacterium. This discrepancy can be attributed to the complex regulatory network of flagellar mediated motility of each bacterial specie that involves different transcriptional factors, signal molecules and/or regulatory small RNAs (sRNAs) (*Mika & Hengge, 2013*).

In *E. coli* and *Salmonella* spp., swimming motility is involved in the adherence of bacteria to host cells and the subsequent colonization (*Wolfson et al., 2020*; *Westerman, McClelland & Elfenbein, 2021*; *de Oliveira Barbosa et al., 2017*). On the other hand, *S. aureus* does not present swimming motility; instead, it performs a surface displacement named colony spreading (*Kaito & Sekimizu, 2007*). This movement is regulated by the *agr* locus, which regulates the expression of toxins and adhesion proteins and is also involved in its virulence (*Kizaki et al., 2016*).

The mode of action of these compounds has not been described. Bacterial motility is a virulence trait tightly regulated, and each bacterial species has its regulatory system (*Chaban, Hughes & Beeby, 2015*). Nevertheless, to carry out this physiological process, a proper function of the cell membrane is required (*Ménard et al., 2014*; *Khan et al., 2017*). In this sense, the impairment of bacterial motility caused by bismuth complexes is attributed to cell membrane damage and further internalization (*Wang et al., 2020*). For this reason, it would be interesting if the bismuth complex obtained in the present study could affect the bacterial membrane functionality.

The production of virulence factors in pathogenic bacteria is regulated in multiple ways; one of them is *quorum-sensing*, a cell-to-cell communication process in which bacteria coordinates its behavior in a population-dependent manner (*Haque et al., 2018*). Figure 8 shown that Bi-edtabz was able to interfere with the production of the *quorum-sensing* regulated pigment (violacein) in the biosensor strain *C. violaceum* that uses AHL as signal

molecules. In pathogenic bacteria such as *P. aeruginosa*, the signal molecule 3-oxo-C12-AHL triggers the expression of several extracellular virulence factors, promotes biofilm maturation, and regulates the expression of antibiotic efflux pumps; thereby, it is considered a key target in the pathogenesis of *P. aeruginosa* (*Lee & Zhang, 2015*; *Tapia-Rodriguez et al., 2019*; *Sarabhai et al., 2016*).

In the other hand, *E. coli* and *Salmonella* spp. does not produce AHL, but they can sense and respond to external AHL produced by other bacteria that influence biofilm formation and the adhesion to epithelial cells during pathogenesis (*Kim et al., 2014*; *Suzuki et al., 2002*; *Smith, Fratamico & Yan, 2011*; *Crago & Koronakis, 1999*). However, *L. monocytogenes* and *S. aureus* use a polypeptide signal molecule instead of the small diffusible molecules used by the other *quorum-sensing* systems (*Novick & Geisinger, 2008*). For this reason, *C. violaceum* model is not useful to relate the treatment effect on these bacteria. This preliminary screening of anti-*quorum sensing* activity showed that bismuth complex could interfere with this cell-to-cell communication process; however, more studies are required to clarify the anti-*quorum sensing* mechanism and its relationship with the biofilm and motility inhibitory activities described here.

## CONCLUSIONS

In this study, a bismuth complex with an open-chained EDTA-based phenylene ligand was synthesized and characterized. [1]H NMR analysis confirmed that the chelation of the bismuth ion to the ligand created a rigid system. Further studies should be made to ensure the coordination sphere of the complex, for example, the X-ray diffraction of a monocrystal of this complex. On the other hand, the pH-variable experiment showed that the complex disassociation occurs as the pH becomes basic and the complex seems to be stable in acid media, which is a good property to consider its use as a pharmaceutical. In addition, the evaluation of anti-virulence properties showed that the bismuth complex and the ligand presented a dose-dependent response and a specific compound-bacteria relationship against biofilm formation, motility, and *quorum sensing*. It is important to highlight that Bi-edtabz presented higher biomass reduction in biofilm formation assay and impaired the motility of *E. coli* O157: H7 and *S. aureus* and inhibited violacein production in the *quorum-sensing* biomonitor strain *C. violaceum*. Thus, edtabz and Bi-edtabz complex can be used as novel anti-virulence agents against pathogenic bacteria.

## ACKNOWLEDGEMENTS

The authors thank the Supramolecular Chemistry Thematic Network and the University of Sonora for facilitating the NMR facilities.

### Funding

This work was supported by the National Council of Science and Technology (CONACYT, Mexico). The funders had no role in study design, data collection and analysis, decision to publish, or preparation of the manuscript.

## Grant Disclosures

The following grant information was disclosed by the authors:
National Council of Science and Technology (CONACYT, Mexico).

## Competing Interests

The authors declare that they have no competing interests.

## Author Contributions

- Melissa Beltran-Torres conceived and designed the experiments, performed the experiments, analyzed the data, prepared figures and/or tables, authored or reviewed drafts of the article, and approved the final draft.
- Rocio Sugich-Miranda conceived and designed the experiments, analyzed the data, prepared figures and/or tables, authored or reviewed drafts of the article, and approved the final draft.
- Hisila Santacruz-Ortega analyzed the data, authored or reviewed drafts of the article, and approved the final draft.
- Karla A. Lopez-Gastelum performed the experiments, analyzed the data, authored or reviewed drafts of the article, and approved the final draft.
- J. Fernando Ayala-Zavala conceived and designed the experiments, analyzed the data, prepared figures and/or tables, authored or reviewed drafts of the article, and approved the final draft.
- Fernando Rocha-Alonzo analyzed the data, authored or reviewed drafts of the article, and approved the final draft.
- Enrique F. Velazquez-Contreras analyzed the data, authored or reviewed drafts of the article, and approved the final draft.
- Francisco J. Vazquez-Armenta conceived and designed the experiments, performed the experiments, analyzed the data, prepared figures and/or tables, authored or reviewed drafts of the article, and approved the final draft.

## Data Availability

In Fig. S1, the molecular weight of the complex was confirmed by ESI-MS. In Fig. S2, the FTIR spectra of the free ligand and the complex were taken with ATR in order to understand the coordination of bismuth with the ligand. In Fig. S3, the full spectrum of the HSQC experiment is given in order to understand the coordination environment of the metal with the ligand and also provide the proper assignment of the protons.

## Supplemental Information

Supplemental information for this article can be found online at http://dx.doi.org/10.7717/peerj-ichem.4#supplemental-information.

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
