# Peer review of "Synthesis and characterization of bismuth (III) complex with an EDTA-based phenylene ligand and its potential as anti-virulence agent"

_PeerJ Inorganic Chemistry, doi:10.7717/peerj-ichem.4_

## Round 0.1 · original submission · Major Revisions

Please provide a comprehensively revised version addressing the editorial comments and a detailed rebuttal letter.

Reviewer 1 ·

Basic reporting

The manuscript has been well-written and all information has been provided in a clear and concise manner. Literature cited is appropriate, albeit some missing, which have been mentioned in the reviewer report. The figures have been properly prepared and captioned accordingly. Overall, the manuscript quality is good with the conclusions arrived to by the authors backed by experimental data.

Experimental design

Experimental data has been reported appropriately, albeit with some missing information, which has been mentioned in the reviewer report. The inference drawn from the experimental data has been presented thoroughly in the manuscript.

Validity of the findings

The overall quality of the manuscript is good. The conclusions arrived to by the authors and thus stated are precise and backed by experimental results.

Additional comments

NA

Annotated reviews are not available for download in order to protect the identity of reviewers who chose to remain anonymous.

Reviewer 2 ·

Basic reporting

no comment

Experimental design

Bismuth can form interactions with oxygen under basic conditions, but under acidic conditions, bismuth nitrate usually behaves as a salt. Bi(NO3)3 5H2O, which I assume is the one that was used, usually releases nitric acid into the reaction medium (and has therefore been described as a nitrating agent) so it is very likely that a complex is formed between edtabz and hydrated Bi(NO3)3 and that the nitrogens form ammonium nitrates; this would explain the high m/z close to 1000 found in the mass spectrum.
There is no consistency between the proposed formula (%. Calcd for BiC22H24O12N6 • 2NO3) and m/z 709.6 (the base peak). Also in this same context it is convenient to review the elemental analysis C, 33.39 %; H, 3.31 %; N, 10.62%.
The integrals of the NMR signals are not shown and the doubt remains as to whether they correspond to the proton in the ortho (2H), meta (2H) and para (1H) positions as suggested. It is convenient to clarify it because the signal assigned as "para" does not seem at first sight to integrate for one proton. There are unassigned signals that appear to be important, such as an apparent doublet at 3.9 ppm with a coupling J similar to the double signal at 4.2 assigned as Hc. It would be convenient to include the complete spectrum with the integrals indicated as supplementary material. I do not consider it necessary to explain the basics of interpretation for this spectroscopy for a specialized reader (HSQC explanation)
"The Hb protons are assigned to the signals at 4.31 and 4.23 ppm. These protons are from the carboxylate group in the acid" I assume that the text refers to the methylene protons neighboring the carboxylate and not to those of the carboxylic acid.
The biological results are very interesting, but it is convenient to use bismuth nitrate and edtabz solutions as a control and rule out that the effect is due to these compounds and not to the bismuth complex.

Validity of the findings

no comment

Additional comments

With more work on the spectroscopic interpretation, I consider that this work can be a nice contribution to the chemistry of bismuth as it is interesting from the chemical perspective as well as its microbiological application.

---

## Round 0.2 · accepted · Accept

Thanks for addressing all the revisions and corrections requested. Now your manuscript is accepted in PeerJ Inorganic Chemistry.

Reviewer 2 ·

Basic reporting

No comment

Experimental design

No comment

Validity of the findings

No comment

Additional comments

The suggestions of the previous version have been considered by the authors and I believe that the article can be published in its present form.